# Effects of chlorpyrifos on the crustacean *Litopenaeus vannamei*

**Edisson Duarte-Restrepo[1,2], Beatriz E. Jaramillo-Colorado[1]\*, Laura Duarte-Jaramillo[1]**

**1** Agrochemical Research Group, Chemical Program, School of Exact and Natural Sciences, University of Cartagena, Cartagena, Colombia, **2** Doctoral Program in Environmental Toxicology, School of Pharmaceutical Sciences, University of Cartagena, Cartagena, Bolivar, Colombia

\* bjaramilloc@unicartagena.edu.co

**Data Availability Statement:** All relevant data are within the manuscript and its Supporting Information files.

**Funding:** Program to Support Research Groups (Res. 2200/2014), sponsored by the Vice-Presidency for Research of the University of

## Abstract

Shrimps can be used as indicators of the quality of aquatic systems exposed to a variety of pollutants. Chlorpyrifos is one of the most common pesticides found in environmental samples. In order to evaluate the effects of chlorpyrifos, adult organisms of *Litopenaeus vannamei* were exposed to two sublethal concentrations of the pesticide (0.7 and 1.3 µg/L) for four days. The $LC_{50}$ (96-hours) value was determined and Lipid oxidation levels (LPO) and the activities of catalase (CAT), glutathion peroxidase (GPx), glutathion-S-transferase (GST) were assessed on the muscle, hepatopancreas and gills from the exposed organisms. In addition, inhibition of acetylcholinesterase (AChE) was determined in the brain. $LC_{50}$ (96-hours) was 2.10 µg/L of chlorpyrifos. Catalase activity and LPO were elevated in the three tissues, whereas a decrease of AChE activities in the brain and an increase of GST activity in the hepatopancreas were observed.

## Introduction

Marine organisms can be used as indicators of the quality of aquatic systems exposed to a variety of environmental pollutants. Invertebrates are exposed to xenobiotics resulting from agricultural fields via the surface runoff of water or indirectly through the trophic chain of the ecosystem [1].

Pesticides play an important role in sustaining agricultural production by protecting crops from pests and vector-borne diseases [2–6]. Organophosphorus (OP) compounds are toxic, and their toxicity mechanism is related to the inhibition of acetylcholinesterase (AChE), resulting in accumulation of acetylcholine in the cholinergic receptors of peripheral and central nervous systems [7].

Chlorpyrifos [O,O-diethyl-O-(3,5,6-trichloro-2-pyridinyl) phosphorothionate] (CAS *Number* 2921-88-2) (**Fig 1**), is an OP insecticide widely used in Colombia for agricultural and domestic pest control, but this also results in non-targeted organisms being exposed to either lethal or sublethal concentrations of this contaminant [8–12].

Crustaceans exposed to OP insecticides have shown an inhibition of AChE, as well as oxidative stress, oxidative metabolism alterations, osmoregulation and immunological responses [11]. Even though AChE inhibition is a widely employed biomarker of OP contamination,

Cartagena, (Grant No 023-2015), as well as CENIACUA (Aquaculture Research Center) Cartagena (Colombia) for the help in collecting shrimps. The funders had no role in study design, data collection and analysis, decision to publish, or preparation of the manuscript.

**Competing interests:** The authors have declared that no competing interests exist.

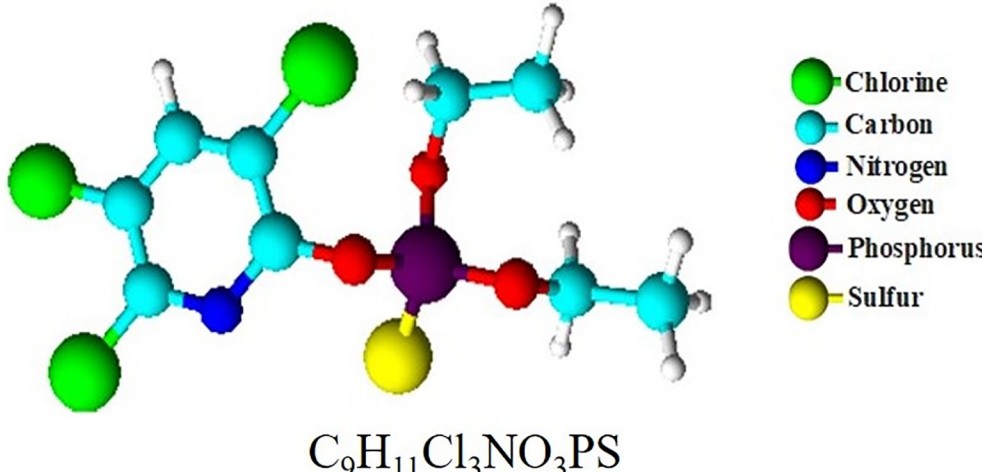

**Fig 1. Chemical structure of chlorpyrifos.**

other oxidative stress-related parameters have been used as indicators of exposure to xenobiotics in aquatic invertebrates as well [12–13].

Once chlorpyrifos reaches target organisms, an oxidative desulfuration of the P = S moiety to P = O occurs, a reaction which is catalyzed by cytochrome P-450 (CYP 1A1) (Phase I), resulting in the toxic intermediate chlorpyrifos-oxon, which can inhibit AChE activity. In additon, chlorpyrifos can undergo dearylation while catalyzed by CYP1A1, resulting in 3,5,6-trichloro-2-pyridinol (TCP), or conjugation by glutathion-S-transferases (GST), sulfo-transferases and glucuronil-transferases to form the corresponding glutathion, glucuronide and sulfate conjugates (detoxification) [9,10]. The pathways of chlorpyrifos in shrimp are shown in **Fig 2**.

The antioxidant defense system includes enzymes such as catalase (CAT), glutathion peroxidase (GPx) and glutathione S-transferase (GST) [14]. In addition, increased lipid peroxidation (LPO) is one of the major contributors to the loss of cell function in oxidative stress situations. CAT is an essential enzyme to promote the degradation of $H_2O_2$, a precursor of the hydroxyl radical that induces DNA damage, protein degradation and lipid peroxidation [14]; GSTs are major phase II-related enzymes, which conjugate electrophilic compounds with reduced glutathion (GSH). For these reasons, OPs are less toxic and more soluble in water, and, therefore, can be quickly excreted from cells after further metabolism [15]. LPO determination has also been successfully employed in aquatic animals to indicate oxidative stress induced by organic contaminants, including OP pesticides [16]. Moyano et al., (2017) reported that chlorpyrifos induces, after acute and long-term exposure, apoptosis and necrosis, partially mediated through AChE overexpression, by using septal SN56 basal forebrain cholinergic neurons [17].

Invertebrates do not have a spinal cord or a spine; instead, most of them have an exoskeleton that spans the entire body. They do not have lungs since they breathe through their skin [18]. The liver participates mainly in the metabolism and regulation of nutrients in aquatic animals and plays an important role in oxidative stress and lipid peroxidation [19]. Crustaceans do not have a liver; they have a hepatopancreas, with functions similar to those of the liver in fish. Therefore, the investigation of possible biochemical changes in response to xenobiotics in crustaceans are done through the hepatopancreas [20]. In the **Fig 3** the anatomy of shrimp (*L. vannmaei*) can be seen. However, an understanding of the influential mechanism underlying the pesticides in the hepatopancreas is still limited and needs further investigation, since most

**Fig 2. Chlorpyrifos biotransformation pathway in invertebrates.** Chlorpyrifos-Oxon; TCP, 3,5,6-trichloro-2-pyridinol; AchE, acetylcholinesterase; CYP 1A1, cytochrome P-450; NADPH, nicotinamide adenine dinucleotide phosphate.

of the research efforts are aimed at biological models of vertebrate organisms [3,4,10], and very few at that of invertebrates, such as *L. vannamei*, which, precisely because of its economic importance and participation in the food supply of humans, must be ensured in terms of food safety.

It has been observed that shrimps are more sensitive to pesticides than fish and mollusks, and have been proposed as indicators of estuarine health due to their worldwide distribution [21]. Thus, *L vannamei* can be used as a test organism for in situ assessment of water contamination [8]. Moreover, crustaceans represent an advantageous tool to monitor environmental contamination as their communities are numerous and can be easily grown under laboratory conditions [18]. Therefore, the aim of this study was: (a) to determine $LC_{50}$ of chlorpyrifos on the *Litopenaeus vannamei* shrimp and (b) assess the effects of sublethal concentrations of chlorpyrifos in this species by using a variety of potential biomarkers including, CAT, GST, GPx, AChE activities, and LPO levels.

## Materials and methods

### Animal collection and maintenance

Shrimps (*Litopenaeus vannamei*) of both genders with an average weight of 48 ± 3 g were collected at CENIACUA (Aquaculture Research Center) located about 20 kilometers (12.4 miles) from Cartagena (Colombia). The animals were transported to the Agrochemical laboratory at University of Cartagena (Colombia) and acclimatized for seven days in 32-liter plastic (polypropylene) tanks containing aerated seawater (30% salinity, pH 8.0, dissolved oxygen between 7.8–9.5 mg/L and 12h/12h dark/light cycles) at 26± 0.6 °C (**S1 Fig**). The shrimps were

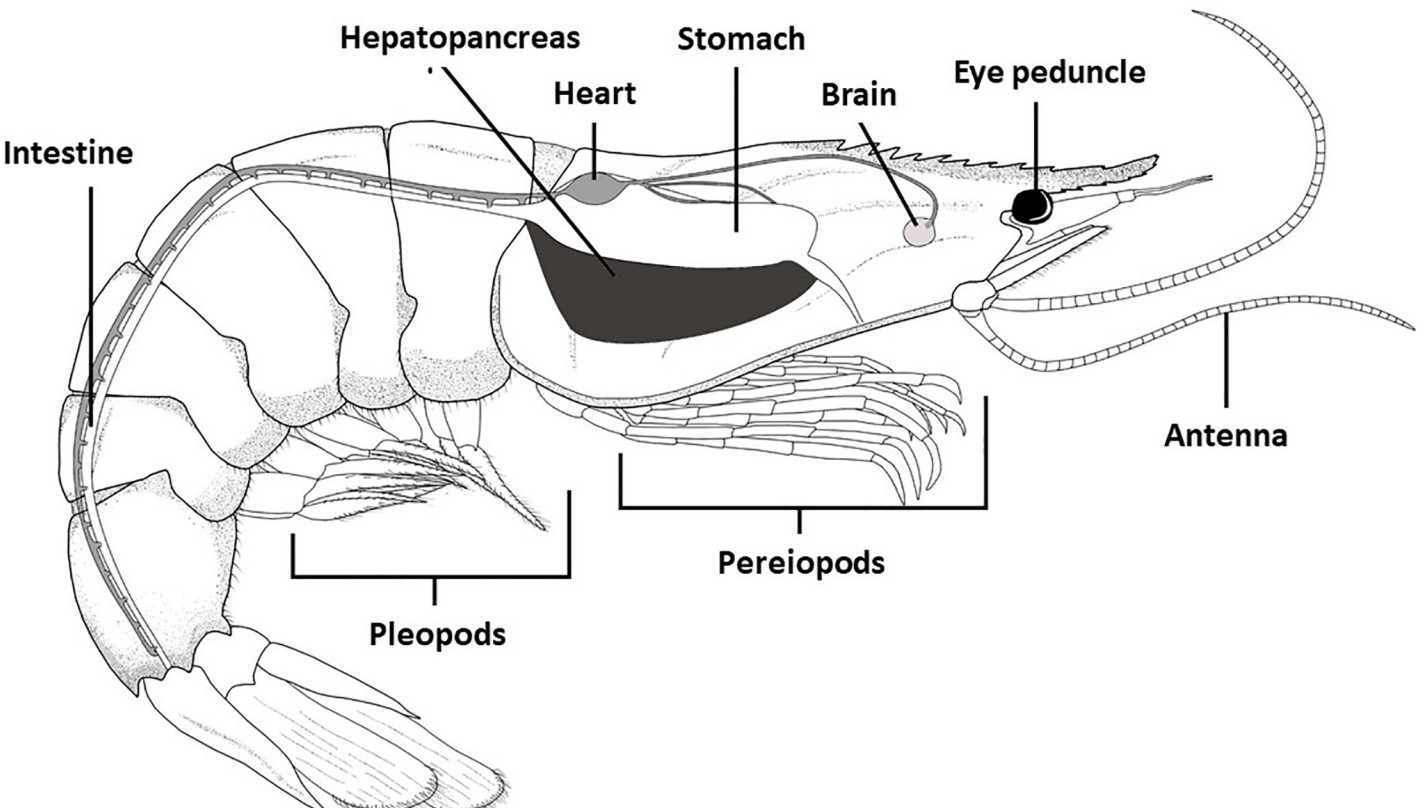

**Fig 3. Anatomy of *Litopenaeus vannamei*.** (Fully owned source; the final illustration was made by using the GIMP Software v2.8.14 (GNU Program for Image Manipulation available at https://www.gimp.org/downloads/).

fed only during acclimation with pellets, formulated shrimp diet (Solla, Colombia), and the water was renewed every two days.

For this study, the gender of *L. vannamei* was not considered. Shrimps were not currently undergoing moults. Experimental procedures were performed in accordance with the principles of the National Animal Protection Status (Colombian law 84 of 1989) established by the University of Cartagena's Central Committee of Ethics in Scientific Research (Resolution 01597 of 2014).

## Toxicity tests

A preliminary analysis was carried out to find the range of chlorpyrifos concentrations. Finally, five concentrations (13.0, 5.4, 2.1, 1.4 and 0.7 μg/L) were used in each toxicity test with three replicates per concentration and one control. The test solutions and control water were renewed daily.

Experiments were carried out in plastic polypropylene tanks (35 L capacity), filled with 30 ppt aerated saline water, pH: 8.0, dissolved oxygen between 7.8–9.5 mg/L and 12h/12h dark/light cycles and 26± 0.6 °C. Eight adult shrimps were used in each replicate. A chlorpyrifos stock solution was prepared by diluting 1 mL Lorsban® 4E in 1 L of distilled water. The specimens were not fed the day before or during toxicity tests. Dead shrimps were counted after 24, 48, 72 and 96 h. The $LC_{50}$ after 96 h was calculated using the probit method by following OECD Guideline 203 [22].

## Sampling and chromatographic analysis

500 mL of each aqueous sample was transferred to separatory funnels. Samples were extracted with 100 mL of dichloromethane (DCM) (three times) and dried using sodium sulphate. The extracts were combined in a 500 mL round bottom flask. Then, 5 mL of isooctane were added in a rotary evaporator. The concentrates were quantitatively transferred to 15 mL centrifuge tubes with two 2 mL of hexane and concentrated by nitrogen evaporation up to 1 mL. Finally, 1 μL was injected into the gas chromatograph coupled to a mass detector (GC-MS) [23].

Chromatography separation was performed using an HP-5 MS capillary column (30 m × 0.25 mm × 0.5μm) (J & W Scientific, USA), and a 7890A gas GC-MS chromatograph system (Agilent Co., Palo Alto, CA, USA) equipped with an *Agilent Technologies* 5975N GC-MS series mass selective detector (MSD). The injection temperature (Agilent 4513A) was 250˚C. The carrier gas was Helium (99.99%). The flow rate was 1 mL/min. The column temperature was held at 120˚C for 1 min and then raised to 300˚C at a rate of 15˚C/min. Transfer interface and ion source temperatures were 230 and 300˚C, respectively. Ions were generated by a 70-eV electron at full scan mode (m/z 40–600).

## Sublethal toxicity assay

The sublethal toxicity of chlorpyrifos was evaluated using biomarkers of oxidative stress (CAT, GPX, and GST activities), lipid peroxidation and AChE inhibition in four tissues (brain, hepatopancreas, gills, and muscle) of *L. vannamei*. Two sublethal concentrations of chlorpyrifos were used (0.7 and 1.4 $\mu g\,L^{-1}$) for 96 hours [24]. No mortality was observed under the experimental conditions. Three replicates were done using five shrimps each.

## Tissue preparation for antioxidant enzymes assays

After 96 h exposure, shrimps of each experimental tank were collected, and brain, hepatopancreas, gills, and muscles were dissected, cleaned and immediately washed with ice-cold saline. Samples were weighed and homogenized in ice-cold Potassium Phosphate Buffer (50mM, pH 7.5, EDTA 60mM) [25] containing protease inhibitor (Sigma P2714) by using a glass homogenizer. Then, the homogenate was centrifuged at 3600 rpm for 15 min at 4˚C [KUBOTA (3700), Japan] and the supernatant corresponding to the post-mitochondrial fraction was used to evaluate enzymatic (CAT, GR, GST) and lipid peroxidation levels (LPO).

## AChE activity

Brains were homogenized at 1:10 w/v with ice-cold Potassium Phosphate Buffer (0.1 M, pH 7.5, EDTA 1 mM, and 0.5% Triton + protease inhibitor). Homogenates were centrifuged at 3600 rpm for 15 min at 4˚C, and AChE activity was measured by using a colorimetric method [26,27]. The reaction was carried out by adding 10 μL of 75 mM substrate of ATCH (acetylthiocholine iodide) into a 990 μL reaction mixture containing 50 μL of 0.01 M dithiobisnitrobenzoate (DTNB) in 0.1 M phosphate buffer (pH 8.0). The optical density increase rate of the reaction medium was measured using a spectrophotometer at 412 nm for 240 s at room temperature. Specific activity is expressed as nmoles of product formed per $min^{-1}\,mg^{-1}$ protein. One AChE unit was the amount of enzyme that hydrolyzed 1 nmol of acetylcholine/min/mg of protein.

## Protein determination

Protein determination was carried out according to the Bradford method [28] with Coomassie Brilliant Blue G-250 by using bovine serum albumin as a standard. The absorbance of samples was measured at 595 nm.

## Catalase (CAT) activity

Catalase activity was determined by following the spectrophotometric decomposition of $H_2O_2$ at 240 nm, in a reaction mixture containing a 50 mM potassium phosphate buffer (pH 7) and 10 mM $H_2O_2$ [29]. Results were expressed as pmol CAT per mg of protein. One CAT unit was the amount of enzyme required to catalyze 1 pmol of $H_2O_2$/min.

## Glutathion-S-transferase (GST) activity

Glutathion-S-transferase (GST) activity was measured using 1-chloro-2,4-dinitrobenzene (CDNB) as substrate, according to the methodology described by Sharbidre [30]. The final reaction mixture contained 1 mM CDNB, and 1 mM reduced glutathion (GSH). The results were expressed as units of GST per mg of protein. One GST unit represented the amount of enzyme required to conjugate GSH with 1 μmol of 1-chloro-2,4-dinitrobenzene/min.

## Glutathion peroxidase (GPX) activity

The glutathion peroxidase (GPX) activity was measured using the method described by Li et al. (2016) [31]. The reaction mixture consisted of a 50 mM phosphate buffer (pH 7), 1 mM EDTA, 1 mM sodium azide, $NaN_3$, 2 U $mL^{-1}$ glutathion reductase (GR), 2 mM reduced glutathion, GSH, 0.2 mM NADPH. After 10 min incubation at 37°C, the overall reaction was initiated by adding 0.5 mM hydrogen peroxide, $H_2O_2$. Oxidation of NADPH was recorded spectrophotometrically at 340 nm for 5 min.

## Lipid oxidation

The production of malondialdehyde (MDA) was assessed by the thiobarbituric acid reactive substances assay TBARS [14, 32]. MDA reacts with thiobarbituric acid (TBA), and the product is read spectrophotometrically at 535 nm. Homogenate was added 1:1 (v:v) to 5% trichloroacetic acid (TCA), and incubated on ice for 15 min. The solution was then mixed at a 2:1 ratio with 0.67% TBA, and centrifuged at 2200 × g at 4°C for 10 min. The whole supernatant was boiled for 10 min and refreshed at room temperature before the absorbance was recorded. A calibration curve with increasing MDA concentrations allowed the calculation of LPO expressed as nmol MDA equiv. $g^{-1}$ tissue.

## Statistical analysis

Data are presented as mean ± Standard deviations (SD). All values were examined for normality (Shapiro-Wilk test) as well as the homogeneity of variances (Levene's test). 96-h $LC_{50}$ (Median lethal concentrations) with 95 percent confidence limits were calculated by using the SPSS software package, version 25 (SPSS Inc. Chicago, IL, USA). One-Way Analysis of variances (ANOVA) was used to identify differences between control and exposed groups, followed by Tukey's post hoc test (S1 File). The level of significance was set at $p < 0.05$.

## Results and discussion

Chlorpyrifos concentrations in water samples were measured by gas chromatography (GC) coupled to a mass spectrometer detector (MS), before and after their application. The quantification limit was 0.1 µg/L and the detection limit was 0.01 µg/L. Linear calibration curves with concentrations ranging from 0.1 to 5 µg/L were employed, with determination coefficients over 0.98. Recoveries of chlorpyrifos from water were in the range of 106–119%.

Fig 4 shows the fragmentogram of chlorpyrifos obtained by GC-MS. The m/z 198 was obtained due to the elimination of–$C_4H_{10}O_2PS$ molecule from a parent ion molecule m/z 350, leading to the formation of structure at m/z 198 corresponding to 3,5,6-trichloro-2-pyridinol (TCP); this is a principal metabolite from chlorpyrifos [33].

### Determination of LC$_{50}$

Percentages of mortality of *L. vannamei* exposed to chlorpyrifos are shown in the Fig 5 (S1 Data). The 96-h acute LC$_{50}$ value determined for *L. vannamei* was 2.10 µg/L of chlorpyrifos

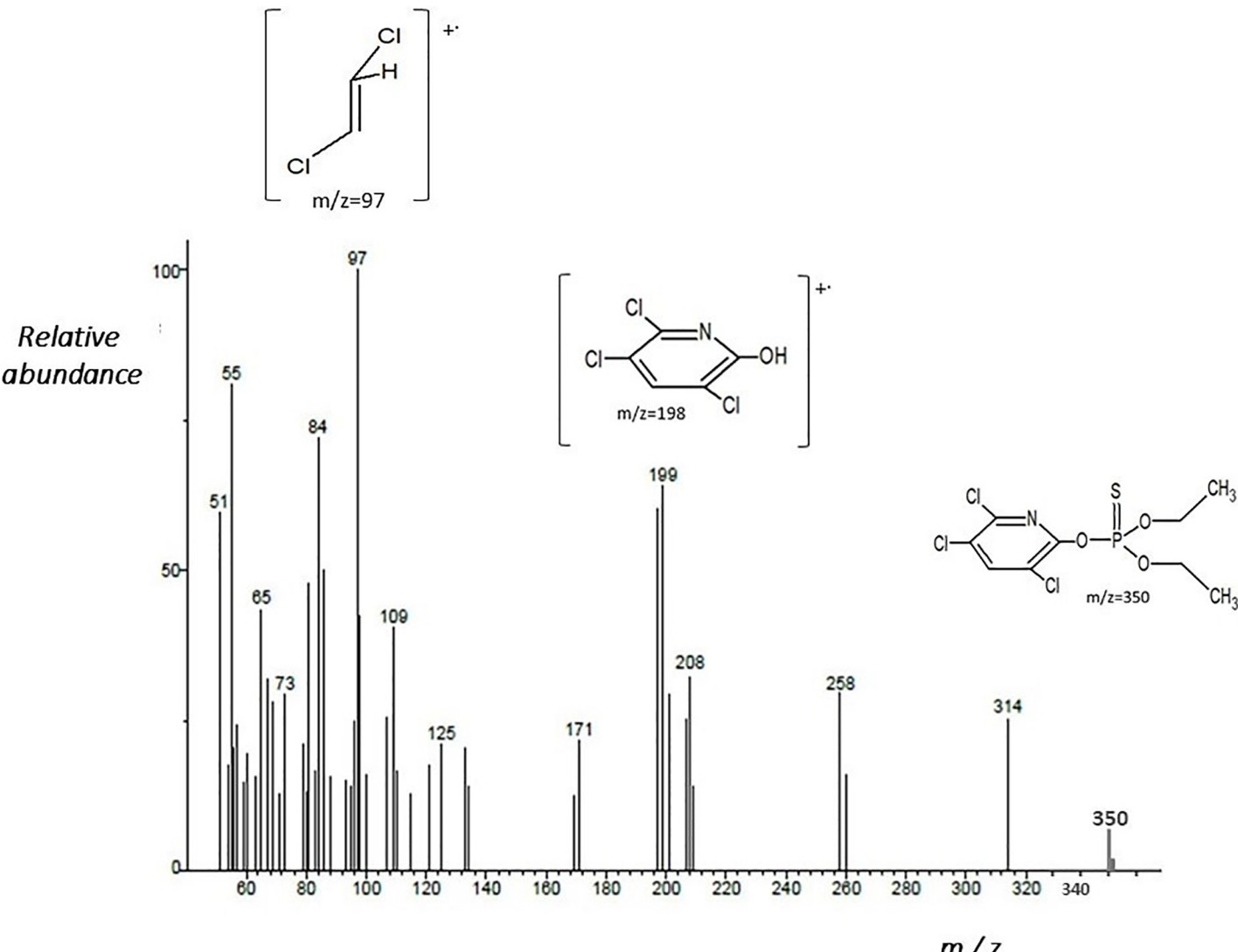

**Fig 4. Fragmentogram of chlorpyrfos obtained by GC-MS.**

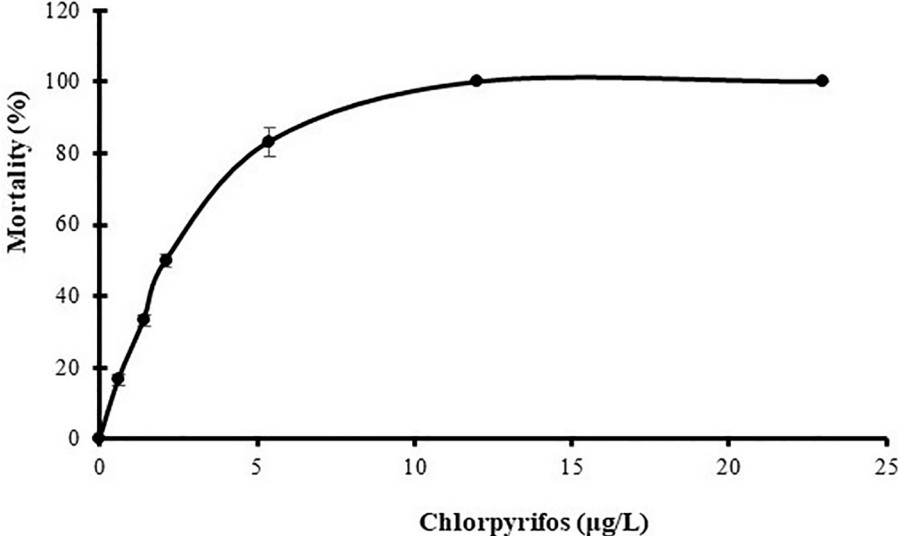

**Fig 5. Percentage of mortality of *L. vannamei* exposed to chlorpyrifos.** Values represent the mean of three determinations+/-SD. (n = 8).

with 95% confidence limits from 2.04 µg/L to 2.16 µg/L. This result allowed for the selection of the pesticide concentrations towards the biomarker analysis. Control mortality was zero. The results are shown in Table 1.

L. vannamei was found to be very sensitive to chlorpyrifos toxicity in comparison with other crustacean species, including *Palaemonetes argentinus* [34], *Paratya australiensis* [35] and *Macrobrachium rosenbergii* [36]. Therefore, it can be concluded that *L. vannamei* is an organism that is sensitive to the exposure to chlorpyrifos, which could serve as a bioindicator of contamination with this pesticide. Thus, the study of the effects of sublethal concentrations of chlorpyrifos on some biomarkers was carried out.

## Acetylcholinesterase inhibition

After four days of sublethal exposure to 0.7 and 1.3 µg/L of chlorpyrifos (**S2 Data**), activity of AChE in brain tissue of *L. vannamei* decreased by 30.8 and 46.2%, respectively (Tukey's test, p<0.05) in a dose-dependent manner compared to the effects observed in the control group

**Table 1. Acute 96-h toxicity of chlorpyrifos in adult shrimp (*Litopenaeus vannamei*).**

| Concentration (µg/L) | Percentage of mortality ± SE | LC$_{50}$ (µg/L) | 95% confidence level | | Regression equation | Chi square |
|---|---|---|---|---|---|---|
| | | | LCL | UCL | | df |
| Control | 0 ± 0 | | | | | |
| 23 | 100 | | | | | |
| 23 | 100 | | | | | |
| 5.4 | 83.3 ± 2.3 | | | | | |
| 2.1 | 50 ± 1.4 | 2.1 | 2.04 | 2.16 | Y = 2.14x + 2.97 | 6.10 |
| 1.4 | 33.3 ± 1.3 | | | | | |
| 0.6 | 16.7± 1.1 | | | | | |

Values of mortality percentages are presented as the overall mean of three replicates ± SE (Standard error). p < 0.05

**Table 2. Activities of AchE enzymes in the brain of *L. vannamei* following four days of exposure to chlorpyrifos.**

| Chlorpyrifos ($\mu g.L^{-1}$) | 0 | 0.7 | 1.3 |
|---|---|---|---|
| AchE (Unit.mg$^{-1}$ protein) | 26±2.3 | 18±2.7 | 14±2.4 |

Values are presented as mean ± S.D. (n = 5).

(**Table 2** and **Fig 6**). There are investigations in invertebrates that associate the presence of OP compounds in the aquatic environment with AChE activity in different tissues [37–40].

Some studies have reported that OP pesticides cause effects through the inhibition of AChE, which leads to an accumulation of the neurotransmitter acetylcholine. In the neuromuscular junction, overstimulation of postsynaptic cholinergic receptors leads to muscle fasciculation and eventual paralysis [41,42].

## Sublethal effects of chlorpyrifos on enzymatic antioxidant defense

The effect of chlorpyrifos on the enzymatic antioxidant defense was evaluated on the hepatopancreas, gills, and muscle tissues of *L. vannamei*. The main defense system of invertebrates is the antioxidant because these organisms have a deficiency of antibodies and acquired immunity. Because of that, the production of high levels of reactive oxygen species (ROS) that generate oxidative stress is considered to be the manifestation of several metabolic pathways resulting in an imbalance of pro-oxidant and antioxidant defense mechanisms [21]. It has been demonstrated that OP compounds can activate oxidative biotransformation processes by generating free radicals and altering antioxidant levels of free radical scavenging enzymes [43, 44].

Oxidative stress as a result of pesticide concentrations has been reported previously [44–48]. Antioxidant enzyme systems have been investigated in aquatic organisms to find biochemical biomarkers that could be used in environmental monitoring systems [48–50]. In this work, alterations in the activity of antioxidant enzymes of *L. vanammei* (CAT, Gpx, and GST)

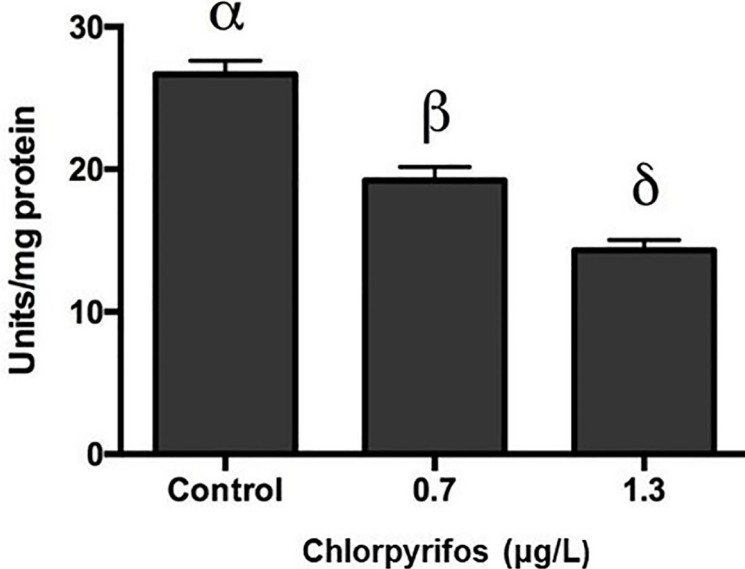

**Fig 6. Effect of chlorpyrifos on Acetylcholinesterase (AchE) activity in *L. vannamei* during sublethal exposure on the brain (0.7 and 1.3 µg L$^{-1}$) for 96 hours.** Data are presented as mean±SD. (p<0.05). (n = 5).

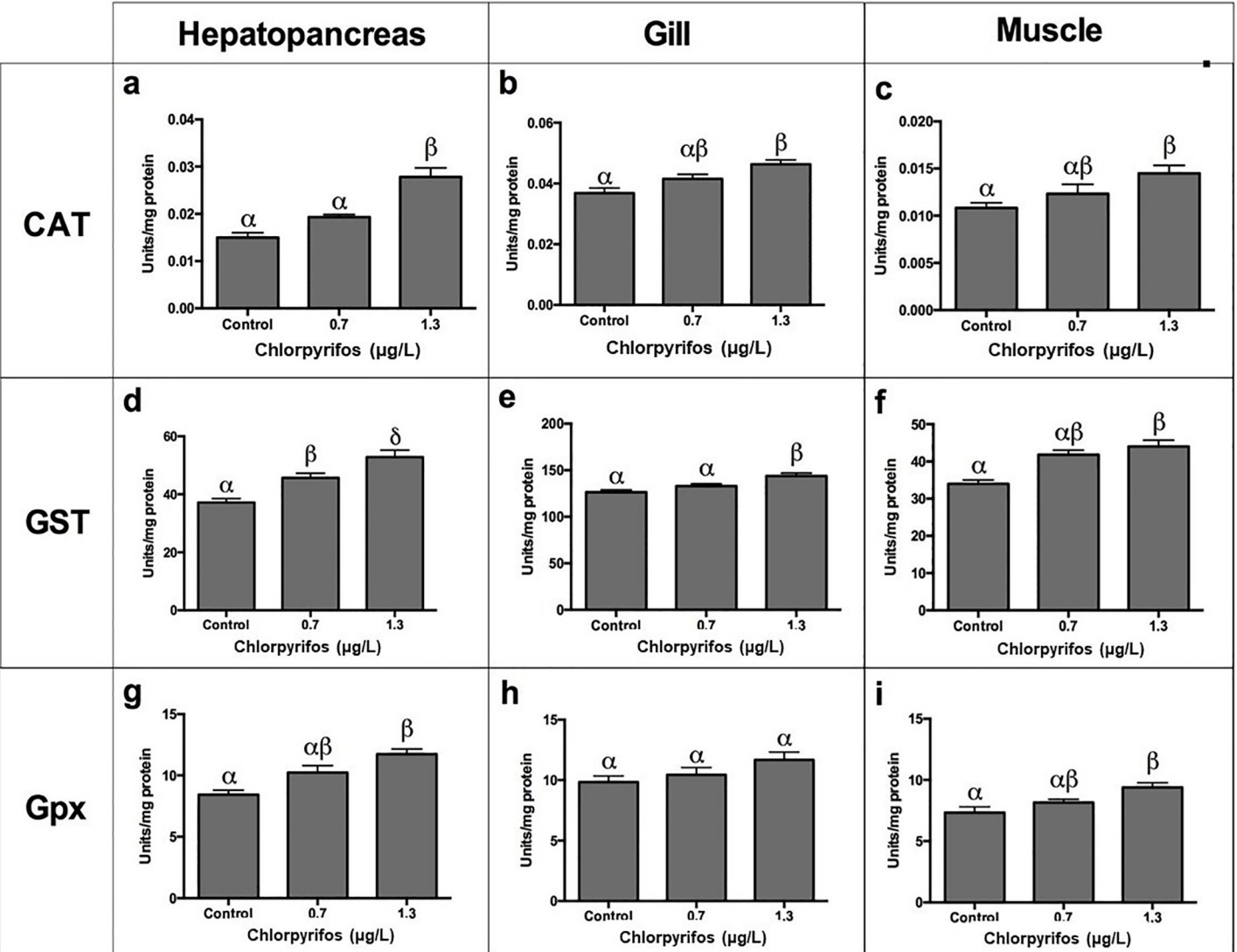

**Fig 7. Effect of chlorpyrifos on CAT, GST, and Gpx activities in heopatopancreas, gills and muscle of *L vannamei*.** Data are presented as mean±SD. (p<0.05). (n = 5).

were found after exposure to two sublethal concentrations of chlorpyrifos, thus suggesting that these changes could be adaptive responses to ROS.

**CAT activity.** After four days of sublethal exposure to 0.7 and 1.3 μg/L of chlorpyrifos, activities of CAT in tissues of *L. vannamei* (muscle, gills, and hepatopancreas) were found to be significantly different from those of the control (Tukey's test, p<0.05). In muscle tissues, CAT activity increased (9.1%, 27.27%). In the gills and hepatopancreas, CAT also showed a dose-dependent elevation (5.12%, 15.38%; and 35%, 100%, respectively) (**Fig 7**).

CAT is an enzyme that regulates high concentrations of $H_2O_2$ [29]. In this research, a significant effect was found in the activity of CAT in the hepatopancreas of *L. vannamei* exposed to chlorpyrifos. In all treatments applied in this study, the action of the CAT enzyme in the hepatopancreas was higher than in the other tissues. Rőszer (2014) showed that mollusks and crustaceans integrate immune functions to a metabolic organ, that is, the mid-intestinal gland

(hepatopancreas) [51]. This is considered to be a gland with high metabolic activity, in which a large production of ROS is expected due to its role as a detoxifying organ [21,51].

Researches have found a dose-dependent increase in the activities of SOD and CAT in diferents tissues of fish exposed to dichlorvos [52], methyl-chlorpyrifos [53], methyl paration [54], fenitrothion [55], and diazinon [56]. Thus, the existence of an inducible antioxidant system may reflect an adaptation of organisms. CAT is reported as one of the key enzymes that condense ROS generated during the bioactivation of xenobiotics in the hepatic tissues [56].

**GST activity.** After four days of exposure to the lowest chlorpyrifos concentration (0.7 μg/L), the activity of GST in muscle, gills and hepatopancreas increased (24.1, 4.75 and 22.53%, respectively), but as the dose increased (1.3 μg/L), a gradual elevation was observed (29.44, 13.47 and 40.42%, respectively). Significant differences were observed when compared to the control (Tukey's test, $p < 0.05$). GST activity in Hepatopancreas was higher than in other tissues (**Fig 7**).

GST is responsible for catalyzing the conjugation of GSH with a wide range of electrophilic substances that could be produced endogenously or by means of lipid peroxidation. This enzyme also has activity against xenobiotics [57].

In this work, we found a marked rise in GST activity compared to the control in all tissues. This suggests the active involvement of this enzyme in detoxification of chlorpyrifos. the hepatopancreas of crustaceans can metabolize chlorpyrifos, which is transformed into oxidized products [53]. Different studies have shown an induction of GST activity into the hepatopancreas. For instance, an increase in GST activity was observed in *Chasmagnathus granulatus* exposed to methyl-parathion [54], and in *Ictalurus nebulosus* exposed to dichlorvos [51]. Similarly, trichlorfon produced an elevation in the activity of this enzyme in *M. rosenbergii* [58].

**GPx activity.** Shrimps were exposed to sublethal concentrations of chlorpyrifos (0.7 and 1.3 μg/L). After four days, GPx activities were evaluated in hepatopancreas, gills, and muscles (Tukey's test, $p < 0.05$) (**Fig 7**). Three tissues showed a higher GPx activity than the control (gills: 6.12 and 18.36%; hepatopancreas: 21.4 and 39.3%; and muscle: 9.5 and 25.7%, respectively). GPx activity in the hepatopancreas reached a high increase (39.2%) and was greater than that of gills (18.6%) and muscle (25.7%) (**Fig 7**).

The GPx enzyme plays a critical role in the antioxidant defense of crustaceans. This is a multifunctional enzyme that prevents the oxidative damage and the formation of lipid hydroperoxides [57]. In this study, the activity of GPx increased in all tissues, which suggests an increase in the mechanisms to decrease the oxidative damage due to exposure to chlorpyrifos. Some studies have also found an elevation in GPx activity by exposing different marine organisms to OP pesticides [59, 60].

## Lipid oxidation

Levels of malondialdehyde (MDA) in *L. vannamei* exposed to two concentrations of chlorpyrifos (0.7 and 1.3 μg/L) showed a significant elevation of lipid peroxidation in the three tissues (muscle, gills, and hepatopancreas) when such results were compared to the control ($p < 0.05$). Muscle tissues showed the highest level of MDA (111.3%) (**Fig 8**).

The peroxidation of polyunsaturated fatty acids (PUFAs) is used as a biomarker of effect [32,49,50]. LPO is considered as the first step of cellular membrane damage by OP pesticides [60]. In this study, a significant increase in the lipid peroxidation of membrane lipids of all assessed tissues (hepatopancreas, gills, and muscles) indicates that ROS-induced damage is one of the main toxic effects of chlorpyrifos (**Fig 8**) [60]. These results are congruent with high levels of lipid peroxidation observed in marine organisms (*Mytilus edulis*, *Gambusia affinis*, *Oreochromis niloticus* and *Poecila reticulate*) exposed to sublethal concentrations of

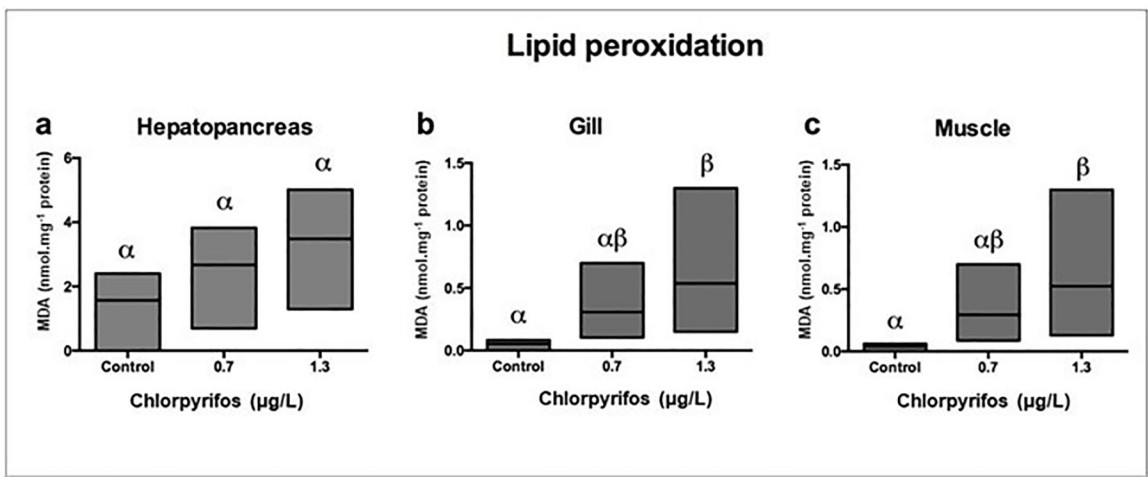

**Fig 8. Effect of chlorpyrifos on LPO in heopatopancreas, gills and muscle of *L vannamei*.** Data are presented as mean±SD. ($p < 0.05$). (n = 5).

chlorpyrifos [60–62]. The observed LPO resulting from exposure to chlorpyrifos may lead to cell death [25]. Hence, *L vannamei* can be used as a test organism for in situ assessment of lipid oxidation.

## Conclusions

This work provides evidence of chlorpyrifos toxicity in *L. vannamei* that is caused by OP-induced oxidative stress, thus indicating that the biological response to sublethal concentrations of the contaminant could be a valuable tool for monitoring OP contamination in freshwater environments. The availability of this robust and economical method sustains the use of *L. vannamei* as a model organism in monitoring studies.

## Supporting information

**S1 Fig. Picture of acclimation of *L. vannamei* in the laboratory.**
(TIF)

**S1 File. File of the statistical treatment of the trials essays, using the SPSS software package, version 25.**
(DOCX)

**S1 Data. Lethal concentration of chlorpyrifos to reach 50% mortality of shrimp within 96 hours (96-h LC50) of *the L. vannamei*.**
(XLSX)

**S2 Data. Dates of enzimatic activities.**
(XLSX)

## Acknowledgments

The authors would like to thank the Research Group Support Program (Res. 2200/2014), sponsored by the University of Cartagena's Vice-Presidency for Research, (Grant No 023–2015), as well as CENIACUA (Aquaculture Research Center) Cartagena (Colombia) for the help in collecting the shrimps.

## Author Contributions

**Conceptualization:** Edisson Duarte-Restrepo, Beatriz E. Jaramillo-Colorado.

**Formal analysis:** Edisson Duarte-Restrepo, Laura Duarte-Jaramillo.

**Funding acquisition:** Beatriz E. Jaramillo-Colorado.

**Investigation:** Edisson Duarte-Restrepo, Beatriz E. Jaramillo-Colorado.

**Methodology:** Edisson Duarte-Restrepo, Laura Duarte-Jaramillo.

**Project administration:** Beatriz E. Jaramillo-Colorado.

**Supervision:** Beatriz E. Jaramillo-Colorado.

**Validation:** Edisson Duarte-Restrepo.

**Writing – original draft:** Edisson Duarte-Restrepo, Beatriz E. Jaramillo-Colorado.

**Writing – review & editing:** Edisson Duarte-Restrepo, Beatriz E. Jaramillo-Colorado.

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
