## [Editor Report · Decision Letter 0]

9 Sep 2019

PONE-D-19-20201

Effects of chlorpyrifos on the crustacean Litopenaeus vannamei

PLOS ONE

Dear Dr. Beatriz E Jaramillo-Colorado,

Thank you for submitting your manuscript to PLOS ONE. After careful consideration, we have decided that your manuscript does not meet our criteria for publication and must therefore be rejected.

Specifically:

The target chemical Chlorpyrifos is a traditional pesticide and its molecular mechanism of toxicity effect is relatively clear, and the study method is also traditional. So I think the work is not innovative on the whole, and it can not be accepted to be published in the journal.

I am sorry that we cannot be more positive on this occasion, but hope that you appreciate the reasons for this decision.

Yours sincerely,

Zhenguang Yan

Academic Editor

PLOS ONE

Additional Editor Comments (if provided):

The work is not innovative, so it can not be accepted.

- - - - -

---

## [Author Response · Author response to Decision Letter 0]

6 Nov 2019

Cartagena of Indias, 6 of November /2019

Staff 

PLOS ONE

Kind regards.

 The authors appreciate the important comments from the academic Editor. In the adjunt file (answer for reviewer), We explain the reasons that justify the publication of this article entitled: Effects of Chlorpyrifos on the crustacean Litopenaeus vannamei.

 . 

 Thanks so much for your valuable collaboration.

Sincerely.

Beatriz E. Jaramillo C., Chem, Ph.D

Chemistry Program, Agrochemical Research Group, University of Cartagena, Cartagena of Indias, Colombia.

The Academic Editor Wrote:

The Academic Editor Wrote:

“The target chemical Chlorpyrifos is a traditional pesticide and its molecular mechanism of toxicity effect is relatively clear, and the study method is also traditional”. 

Answer: 

We will write three reasons why the manuscript should be published

1) Several research papers can be found on the effect of chlorpyrifos on mammals and fish, but much remains to be studied in aquatic species, especially in invertebrates (crustaceans). For example, when reviewing the topic with keywords such as chlorpyrifos and fish in www.pubmed.com, more than 400 articles appear, while searching in the same database with the keywords: “Litopenaeus vannamei” and “organophosphorus” and “chlorpyrifos” shows one result. We did a bibliographic tracking in the following databases: Sciencedirect, ACS, Scielo and Springer on Chlorpyrifos toxicity and we have found (see table 1 in the adjunt file of answer to reviewer)

Table 1. Articles found in the databases: Sciencedirect, ACS, Scielo and Springer on Chlorpyrifos toxicity

On invertebrates only exist 0,76% of public articles. 

In the Stanford Encyclopedia of Philosophy we read: 

“In developing ideas about the overall value of biodiversity it has been natural to draw on existing arguments about values of individual species (for review, see World Conservation Union 1980; Norton 1988). Commodity value and other direct use values have intuitive appeal because they reflect known values. But a key problem is that species need to be preserved for reasons other than any known value as resources for human use (Sober 1986). Callicott (1986) discusses philosophical arguments regarding non-utilitarian value and concludes that there is no easy argument to be made except a moral one. Species have some “intrinsic value” — reflecting the idea that a species has a value “in and for itself” (Callicott 1986, p.140) — and there is an ethical obligation to protect biodiversity”.

2) There are differences between the physiology of vertebrate and invertebrate organisms. In our study, the target organism was an invertebrate, crustacean, the Litopenaeus vannamei. See in the manuscript pages 3-4

Invertebrates do not have a spinal cord or spine; instead, most of them have an exoskeleton that spans the entire body. They do not have lungs since they breathe through their skin [1]. The liver participates mainly in the metabolism and regulation of nutrients in aquatic animals and plays an important role in oxidative stress and lipid peroxidation [2]. The crustaceans do not have liver; they have hepatopancreas, with functions similar to those of the fish liver (see fig1). Therefore, the investigation of possible biochemical changes in response to xenobiotics in crustaceans are done through the hepatopancreas [3], (see Fig. 1). However, the understanding of the influential mechanism underlying the pesticides in the hepatopancreas is still limited and needs further investigation [4]. 

So in Shrimp doesn’t exist a molecular mechanism of toxicity effect relatively clear.

Is also true that the vast majority of research is directed to biological models of vertebrate organisms, and less to invertebrates, such as L. vannamei, which precisely because of its economic importance and participation in the food of humans, must be ensured their food safety.

¿How chlorpyrifos affects a crustacean as the Litopenaeus vannamei?; what biomarkers could be used to assess its toxic effect

This research contributes to the study of the toxicity of chlorpyrifos in crustaceans, using the shrimp (Litopenaeus vannamei) as a model.

• On the other hand, the cultivation of Shrimp, Litopenaeus vannamei has an important role in economic growth, employment and welfare of coastal communities, around the world. But is very sensible to xenobiotics like pesticides.

 Fig 1. Anatomy of L. vannamei (http://www.parasitosypatogenos.com.ar).

1. Robalino J, Wilkins B, Bracken-Grissom HD, Chan T-Y, O’Leary MA The Origin of Large-Bodied Shrimp that Dominate Modern Global Aquaculture. PLoS One. 2016; 11(7): e0158840. 

2. Lu Q, Sun Y, Ares I, Anadon A, Martinez M, Martínez-Larrañaga MR. Deltamethrin toxicity: A review of oxidative stress and metabolism. Environ. Res. 2019; 170: 260–281.

3. Silveira de Melo M, Gonçalves dos Santos TP, Jaramillo M, Nezzi L, Rauh Muller YM, Nazari EM. Histopathological and ultrastructural indices for the assessment of glyphosate-based herbicide cytotoxicity in decapod crustacean hepatopancreas. Aquat. toxicol. 2019; 210:207-214.

4. Yang Z, Zhang Y, Jiang Y, Zhu F, Zeng L, Wang Y, et al. Transcriptional responses in the hepatopancreas of Eriocheir sinensis exposed to deltamethrin. PLoS ONE. 2017; 12(9): e0184581. 

3) Chlorpyrifos is one of the most utilized insecticides in Colombia for agricultural and domestic pest control, but this also results in non-target organisms exposed to either lethal or sublethal concentrations of this contaminant (García-de la Parra et al., 2006). In Colombia is permit the use the chlorpyrifos, here don’t exist none prohibition. We want to change Colombian lays about concentration limits chlorpyrifos. But politicians think that in Colombia, there is not enough research on this pesticide In Colombia, we need to disseminate research on the effects of chlorpyrifos to pressure government agencies and thus transform the laws that govern the use of this pesticide. With this work we would like to contribute to that objective. (See fig 2 and fig 3, in the adjunt file of answer to reviewer) 

Fig 2. Pesticide production in Colombia. See it in the adjunt file of answer to reviewer

Fig 3. Sales of Chlorpyrifos in Colombia. See it in the adjunt file of answer to reviewer

https://www.ica.gov.co/areas/agricola/servicios/regulacion-y-control-de-plaguicidas-quimicos/estadisticas/boletinplaguicidas2015-12-12-2016.aspx

This work is addressing a problem that could be considered at the regional level, but in reality, it is a global problem. The Bay of Cartagena and its biota has been exposed to a chlorpyrifos spill, and some companies (Dow Agrosciencies, Monsanto (Syngenta) produce this compound even in its industrial zone. See fig 4. See it in the adjunt file of answer to reviewer

Fig 4. Chlorpyrifos spill in Cartagena (Colombia). See it in the adjunt file of answer to reviewer

Fig 5. Plaguicides factories in Cartagena Bay (Colombia). See it in the adjunt file of answer to reviewer

This factories to this day continue to pollute the marine environmental of Cartagena bay with Chlorpirifos (LORSBAN)

Year article

1991 Cowgill RT, Gowland CA, Ramirez V. The history of a chlorpyrifos spill:

Cartagena, Colombia. Environ. Int. 1991; 17: 61-71.

2013 https://sostenibilidad.semana.com/actualidad/articulo/cartagena-dano-ambiental-petroleo-dow-quimica/30230

2015 https://www.eltiempo.com/archivo/documento/CMS-15869078

2015 https://www.javerianacali.edu.co/sites/ujc/files/node/field-documents/field_document_file/caso_presentado_por_estudiantes_derecho_javeriana_cali_en_buenos_aires.pdf

Agrochemical Research Group, University of Cartagena, did a work where sampled marine sediments in nine sites in the Bay of Cartagena in collaboration with the University of Buffalo (New York) and the following concentrations of pesticides were found:

Fig 6. Pesticides concentration in Cartagena Bay. See it in the adjunt file of answer to reviewer

The samples were conducted in June 2015 and 2017. We can see in the figure 6 that in seven of the sampling sites the highest concentration pesticide is chlorpiryfos.

---

## [Decision Letter · Decision Letter 1]

2 Jan 2020

PONE-D-19-20201R1

Effects of chlorpyrifos on the crustacean Litopenaeus vannamei

PLOS ONE

Dear Dr Jaramillo-Colorado,

Thank you for submitting your manuscript to PLOS ONE. After careful consideration, we feel that it has merit but does not fully meet PLOS ONE’s publication criteria as it currently stands. Therefore, we invite you to submit a revised version of the manuscript that addresses the points raised during the review process.

We have received two referee reports ad both have pointed out the need for a major revision to address several issues raised. Kindly respond to each of referee comments.

We would appreciate receiving your revised manuscript by Feb 16 2020 11:59PM. To enhance the reproducibility of your results, we recommend that if applicable you deposit your laboratory protocols in protocols.io, where a protocol can be assigned its own identifier (DOI) such that it can be cited independently in the future. For instructions see: http://journals.plos.org/plosone/s/submission-guidelines#loc-laboratory-protocols

We look forward to receiving your revised manuscript.

Kind regards,

Iddya Karunasagar and Xinghui Qiu

Academic Editors

PLOS ONE

Journal Requirements:

2) We note that Figure(s) [Anatomy of L. vannamei (http://www.parasitosypatogenos.com.ar)] in your submission contain copyrighted images. All PLOS content is published under the Creative Commons Attribution License (CC BY 4.0), which means that the manuscript, images, and Supporting Information files will be freely available online, and any third party is permitted to access, download, copy, distribute, and use these materials in any way, even commercially, with proper attribution. For more information, see our copyright guidelines: http://journals.plos.org/plosone/s/licenses-and-copyright.

a). You may seek permission from the original copyright holder of Figure(s) [Anatomy of L. vannamei (http://www.parasitosypatogenos.com.ar)] to publish the content specifically under the CC BY 4.0 license.

b). If you are unable to obtain permission from the original copyright holder to publish these figures under the CC BY 4.0 license or if the copyright holder’s requirements are incompatible with the CC BY 4.0 license, please either i) remove the figure or ii) supply a replacement figure that complies with the CC BY 4.0 license. Please check copyright information on all replacement figures and update the figure caption with source information. If applicable, please specify in the figure caption text when a figure is similar but not identical to the original image and is therefore for illustrative purposes only.

3) Thank you for your ethics statement:

Experimental procedures were performed in accordance with the principles of animal care

established by the Central Committee of Ethics in Scientific Research of the University of

Cartagena (Act No 80/2015).

a) Please amend your current ethics statement to confirm that your named ethics committee Institutional Care and Use Committee (IACUC) specifically approved this study.

For additional information about PLOS ONE ethical requirements for animal research, please refer to http://journals.plos.org/plosone/s/submission-guidelines#loc-animal-research

4) Thank you for updating your data availability statement. You note that your data are available within the Supporting Information files, but no such files have been included with your submission. At this time we ask that you please upload your minimal data set as a Supporting Information file, or to a public repository such as Figshare or Dryad.

Please also ensure that when you upload your file you include separate captions for your supplementary files at the end of your manuscript.

As soon as you confirm the location of the data underlying your findings, we will be able to proceed with the review of your submission.

Additional Editor Comments (if provided):

We have received two referee reports ad both have pointed out the need for a major revision to address several issues raised. Kindly respond to each of referee comments.

Reviewers' comments:

Reviewer's Responses to Questions

**Comments to the Author**

1. If the authors have adequately addressed your comments raised in a previous round of review and you feel that this manuscript is now acceptable for publication, you may indicate that here to bypass the “Comments to the Author” section, enter your conflict of interest statement in the “Confidential to Editor” section, and submit your "Accept" recommendation.

Reviewer #1: (No Response)

Reviewer #2: (No Response)

2. Is the manuscript technically sound, and do the data support the conclusions?

Reviewer #1: Partly

Reviewer #2: Yes

3. Has the statistical analysis been performed appropriately and rigorously? 

Reviewer #1: Yes

Reviewer #2: Yes

4. Have the authors made all data underlying the findings in their manuscript fully available?

Reviewer #1: No

Reviewer #2: Yes

5. Is the manuscript presented in an intelligible fashion and written in standard English?

Reviewer #1: No

Reviewer #2: Yes

6. Review Comments to the Author

Reviewer #1: 1. The authors need to have the manuscript checked for English usage by someone whose first language is English. Throughout the paper there are many instances of improper English (unnecessarily capitalized words, missing words, misspelled words).

2. The manuscript as downloaded was difficult to review as there were 2 versions that were not well defined. I reviewed the one that had small type at the top that read "Revised Manuscript with Track Changes." There were no figures included with that, so I reviewed the figures associated with another version labeled "Manuscript". Then there were other figures located elsewhere in the download. This was all very confusing.

3. Figures legends were all placed oddly throughout the manuscript and do a very poor job of describing the figures.

4. Table 1 has values that do not match the values in the manuscript.

5. In the methods, it is not clear how many shrimp were placed in each replicate.

6. Reference #25 has the incorrect year.

7. I could not find if the authors made all data underlying the findings in their manuscript fully available. That does not mean it is not there, I just do not know where to find it.

8. I think the manuscript is worth publishing whenever it is satisfactorily revised.

Reviewer #2: Line 38: Last line in this paragraph seems inappropriately placed as no information on oxidative stress has been addressed at this point. Remove or re-word.

Line 102: Was shrimp sex determined or was sex not considered during selection of organisms? This is unclear.

Line 107: More detail on the brand or content of the shrimp feed may be useful.

Line 108: There are many stages of intermoult in a shrimp's life. Clarify if the shrimp were at a certain life stage or state that they were not currently undergoing moult. "The intermoult stage" is unclear.

Line 119: What type of plastic? Could this plastic affect your results at the temperature you held the shrimp at?

Line 218: Why were calibration curves so high (100-5000ug/L)? Test concentrations were only 0.7-13.0 ug/L. Clarify.

Line 229: LC50 value in this line (2.10 ug/L) differs from the LC50 value in Table 1 (0.00021 ug/L). Confirm units.

Line 268: Behaviours are mentioned but observation of behaviour is not outlined in methods.

Line 303: Was chlorpyrifos detected in muscle tissue? This is worded poorly and should be clarified. If stating a dose-dependent relationship then re-word the statement. If indeed the chlorpyrifos content of the muscle increased, include in methods how this was measured.

7. PLOS authors have the option to publish the peer review history of their article (what does this mean?). If published, this will include your full peer review and any attached files.

Reviewer #1: No

Reviewer #2: No

---

## [Author Response · Author response to Decision Letter 1]

25 Feb 2020

Cartagena of Indias, 16 February 2020

Sir/Madam

Editor

PLOS ONE

Kind regards.

I´m sending you the answers to the reviewers, and corrections of the manuscript titled:” Effects of chlorpyrifos on the crustacean Litopenaeus vannamei”. 

 The authors appreciate the important comments from the reviewers. Below, we explain the corrections and changes made to the article. The changes are indicated in the manuscript with red letters.

Thanks so much for your valuable collaboration.

Sincerely.

Beatriz Eugenia Jaramillo-Colorado., Chem, Ph.D

Chemistry Program, Agrochemical Research Group, University of Cartagena, Cartagena of Indias, Colombia.

The Academic Editors Wrote:

ANSWER

It was checked and corrected.

2. We note that Figure(s) [Anatomy of L. vannamei (http://www.parasitosypatogenos.com.ar)] in your submission contain copyrighted images. (1) present written permission from the copyright holder to publish these figures specifically under the CC BY 4.0 license, or (2) remove the figures from your submission:

ANSWER 

That was changed. We made our own illustration, using the GIMP Software v2.8.14.

See Figure 3.

3. Please amend your current ethics statement to confirm that your named ethics committee Institutional Care and Use Committee (IACUC) specifically approved this study.

ANSWER

The files will be loaded in the submit

4. Thank you for updating your data availability statement. You note that your data are available within the Supporting Information files, but no such files have been included with your submission. At this time we ask that you please upload your minimal data set as a Supporting Information file, or to a public repository such as Figshare or Dryad.

Please also ensure that when you upload your file you include separate captions for your supplementary files at the end of your manuscript.

ANSWER

The Supporting information was added to the manuscript, and the files will be upload in the submission.

REVIEWERS´ COMMENTS 

1. Have the authors made all data underlying the findings in their manuscript fully available?

ANSWER.

 The Supporting information will be upload in the submission.

2. Is the manuscript presented in an intelligible fashion and written in standard English?

ANSWER 

The manuscript was checked and corrected by an English language professional. 

 Reviewer #1:

3. Figures legends were all placed oddly throughout the manuscript and do a very poor job of describing the figures.

ANSWER

This was reviewed and corrected throughout the manuscript.

4. Table 1 has values that do not match the values in the manuscript.

ANSWER

It was our mistake. Since we did not consider a unit conversión. This was corrected. See Table 1, and lines 235 to 242, page 10. 

5. In the methods, it is not clear how many shrimp were placed in each replicate.

ANSWER

It was corrected. In toxicity tests, eight adult shrimps were used in each replicate. Please, to see page 5 line 124. In Sublethal toxicity assay Three replicates were done using five shrimps each. See page 6 line 151.

6. Reference #25 has the incorrect year.

ANSWER.

It was corrected. The year is 2015. See page 19, line 481

7. I could not find if the authors made all data underlying the findings in their manuscript fully available. That does not mean it is not there, I just do not know where to find it.

ANSWER.

The Supporting information was added to the manuscript, and the files will be upload in the submission. See page 15.

Reviewer #2

8. I think the manuscript is worth publishing whenever it is satisfactorily revised.

ANSWER

Thanks so much. Below, I describe the changes done. 

8. Line 38: Last line in this paragraph seems inappropriately placed as no information on oxidative stress has been addressed at this point. Remove or re-word.

9. ANSWER.

It was removed.

10. Line 102: Was shrimp sex determined or was sex not considered during selection of organisms? This is unclear.

ANSWER

For this study, the gender of L. vannamei was not considerated. See page page 5 line 110.

10. Line 107: More detail on the brand or content of the shrimp feed may be useful.

ANSWER.

The feed was shrimp diet (Solla, Colombia). See Page 5, line 108

11. Line 108: There are many stages of intermoult in a shrimp's life. Clarify if the shrimp were at a certain life stage or state that they were not currently undergoing moult. "The intermoult stage" is unclear.

ANSWER

Shrimps were not currently undergoing moults. See Page 5, line 111.

11. Line 119: What type of plastic? Could this plastic affect your results at the temperature you held the shrimp at? 

ANSWER. 

The plastic was polypropylene. See page 5, line 106. It does not affect the experiment, since it was carried out at room temprature. 

12. Line 218: Why were calibration curves so high (100-5000ug/L)? Test concentrations were only 0.7-13.0 ug/L. Clarify.

ANSWER

It was our mistake. Since we did not consider a unit conversión. This was corrected. See lines 219- 223, page 9.

13. Line 229: LC50 value in this line (2.10 ug/L) differs from the LC50 value in Table 1 (0.00021 ug/L). Confirm units.

ANSWER

It was our mistake. Since we did not consider a unit conversión. This was corrected. See Table 1, and lines 235 to 255, page 10. 

14. Line 268: Behaviours are mentioned but observation of behaviour is not outlined in methods.

ANSWER.

It was removed.

15. Line 303: Was chlorpyrifos detected in muscle tissue? This is worded poorly and should be clarified. If stating a dose-dependent relationship then re-word the statement. If indeed the chlorpyrifos content of the muscle increased, include in methods how this was measured.

ANSWER 

That was misspelled. Chlorpyrifos was not determined in the muscle. What was evaluated was the activity of catalase in the muscle. See page 12 line 305.

---

## [Decision Letter · Decision Letter 2]

23 Mar 2020

Effects of chlorpyrifos on the crustacean Litopenaeus vannamei

PONE-D-19-20201R2

Dear Dr. Jaramillo-Colorado,

We are pleased to inform you that your manuscript has been judged scientifically suitable for publication and will be formally accepted for publication once it complies with all outstanding technical requirements.

With kind regards,

Iddya Karunasagar

Academic Editor

PLOS ONE

Additional Editor Comments (optional):

All comments addressed satisfactorily

Reviewers' comments:

Reviewer's Responses to Questions

**Comments to the Author**

1. If the authors have adequately addressed your comments raised in a previous round of review and you feel that this manuscript is now acceptable for publication, you may indicate that here to bypass the “Comments to the Author” section, enter your conflict of interest statement in the “Confidential to Editor” section, and submit your "Accept" recommendation.

Reviewer #1: All comments have been addressed

Reviewer #2: All comments have been addressed

2. Is the manuscript technically sound, and do the data support the conclusions?

Reviewer #1: Yes

Reviewer #2: Yes

3. Has the statistical analysis been performed appropriately and rigorously? 

Reviewer #1: Yes

Reviewer #2: Yes

4. Have the authors made all data underlying the findings in their manuscript fully available?

Reviewer #1: Yes

Reviewer #2: Yes

5. Is the manuscript presented in an intelligible fashion and written in standard English?

Reviewer #1: Yes

Reviewer #2: Yes

6. Review Comments to the Author

Reviewer #1: All the previous comments seem to have been addressed. I have no further comments to make to the authors.

Reviewer #2: (No Response)

7. PLOS authors have the option to publish the peer review history of their article (what does this mean?). If published, this will include your full peer review and any attached files.

Reviewer #1: No

Reviewer #2: No

---

## [Editor Report · Acceptance letter]

30 Mar 2020

PONE-D-19-20201R2 

Effects of chlorpyrifos on the crustacean *Litopenaeus vannamei*

Dear Dr. Jaramillo-Colorado:

I am pleased to inform you that your manuscript has been deemed suitable for publication in PLOS ONE. Congratulations! Your manuscript is now with our production department. 

With kind regards,

on behalf of

Dr. Iddya Karunasagar 

Academic Editor

PLOS ONE